# Accurate Classification of Dysplasia in Inflammatory Bowel Disease Patients Using Deep Learning

Ahmad Tamim Hamad
*Electrical Eng. & Computer Science*
*The University of Missouri*
Columbia, USA
ahzkc@missouri.edu

Parshad Suthar
*Electrical Eng. & Computer Science*
*The University of Missouri*
Columbia, USA
phs2dm@missouri.edu

Katsiaryna Laziuk
*HCA Healthcare Pathology Services*
*Research Medical Center*
Kansas City, USA
Katsiaryna.Laziuk@HCAHealthcare.com

Deepthi Rao
*Pathology & Anatomical Sciences*
*The University of Missouri*
Columbia, USA
raods@health.missouri.edu

Praveen Rao
*Electrical Eng. & Computer Science*
*The University of Missouri*
Columbia, USA
praveen.rao@missouri.edu

*Abstract*—Patients with inflammatory bowel disease (IBD) are at high risk for developing dysplasia and colorectal cancer. The early and accurate detection and treatment of dysplasia forms the main strategy to reduce mortality from colorectal cancer in IBD patients. Detecting such dysplasia is challenging because of the subtle, unconventional, multi-focal nature of the lesions. In this work, we develop an approach for accurate classification of dysplasia in IBD patients using Bayesian deep learning. We modify existing deep learning models to perform Bayesian approximation for achieving higher classification accuracy than a deterministic deep learning model. Specifically, we propose to insert one or more densely connected layers before the final densely connected layer of a model that performs classification. Each newly inserted layer is followed by a dropout layer. These inserted dropout layers are enabled during training and inference. Instead of obtaining a single prediction by a deterministic model for a given test input, we obtain a distribution of predictions and then compute the most probable prediction. We evaluated our approach using 60+ digital slides of histopathology tissue sections containing three different types of dysplasia in IBD patients. Our best Bayesian deep learning model achieved an accuracy of 97.37%, 93.23%, and 98.16%, respectively for the three dysplasia types using patch-wise classification.

*Index Terms*—Dysplasia classification, inflammatory bowel disease, deep learning, whole slide imaging

## I. INTRODUCTION

INFLAMMATORY bowel disease (IBD) encompasses chronic inflammatory states of the gastrointestinal tract. Every year, 70,000 new IBD cases are diagnosed in the United States [1]. The healthcare financial burden of IBD is $31 billion annually [1]. Patients with IBD are at high risk for developing dysplasia, which indicates the presence of abnormal cells [1]. It is reported that about 18% of the IBD patients may develop colorectal cancer by the time they have had IBD for 30 years [2]. It is also known that about 10-15% of IBD patients die of colorectal cancer annually [3]. Hence, accurate assessment and treatment of IBD patients is crucial for prevention of colorectal cancer in these patients.

The process of cancer formation in IBD is more of a non-polypoid mucosal dysplasia, which leads to invasive cancer at an exaggerated rate. For a pathologist, detecting dysplasia is challenging because of the subtle, unconventional, multi-focal nature of the lesions that are located among inflammatory pseudopolyps or scarred post-inflammatory background mucosa [4]. (Even personal communications with expert gastrointestinal pathologists confirmed this situation.) However, with the identification of new types of nonconventional dysplasia [4], [5], the detection of IBD-associated dysplasia has become significantly more challenging for pathologists. If dysplasia is missed or incorrectly graded, it can eventually lead to colorectal cancer in IBD patients. Therefore, *there is a clear need for early and accurate detection of IBD-associated dysplasia by pathologists*. This is regarded as the standard of care to minimize mortality due to colorectal cancer in IBD patients.

Deep learning (DL), a subfield of artificial intelligence (AI), has received much attention in biomedical image analysis [6]–[8] and in diagnostic fields such as pathology [9]–[12]. For pattern recognition in images, a deep neural network learns multiple representations of the input images at different levels of abstractions [13]. It can learn complex, non-linear decision boundaries for achieving high accuracy in classification tasks. In addition, it avoids the tedious process of hand-engineered feature selection required by conventional machine learning techniques. Open-source frameworks such as TensorFlow/Keras[1][2] and PyTorch[3] have commoditized DL. They support a variety of models based on convolutional neural

---

[1] https://www.tensorflow.org  [2] https://keras.io  [3] https://pytorch.org

networks (CNNs) [13] for image classification. These include ResNet [14], DenseNet [15], and EfficientNet [16]. More recently, these frameworks also support Vision Transformer (ViT) [17] that employs the groundbreaking Transformer architecture [18], which has revolutionized the field of natural language processing (NLP).

In this work, we investigate how popular DL models (e.g., DenseNet, EfficientNet, ViT) can be employed for dysplasia classification in IBD patients using whole slide images (WSIs) of histopathology tissue slides. Specifically, we explore if Bayesian DL [19], which uses a Bayesian approach for model training and inference, can provide improved classification performance than a deterministic DL model while capturing the model uncertainty. The key contributions of this paper are as follows:

- We develop a DL-based approach for accurate classification of dysplasia in IBD patients. We specifically focus on three types of dysplasia of interest to a pathologist diagnosing the tissue sections of IBD patients. Given an input patch of a WSI, our approach predicts the type of dysplasia in the patch.
- We employ Bayesian DL to achieve higher classification accuracy and capture the model uncertainty using the idea of dropout for Bayesian approximation [20]. Given a DL model, we insert one or more densely connected layers before the final densely connected layer of the model that performs classification. Each newly inserted densely connected layer is immediately followed by a dropout layer. These inserted dropout layers are enabled during training and inference. As a result, we obtain a distribution of predictions for a test input and then compute the most probable class label for it.
- We evaluated our approach on popular DL models, namely, DenseNet, EfficientNet, and ViT using 60+ WSIs of histopathology tissue slides. We observed that the Bayesian DL models achieved better performance than their deterministic counterparts. Our best Bayesian DL model (based on EfficientNet) achieved 97.37%, 93.23%, and 98.16% accuracy for the three classes using patchwise classification of dysplasia in these slides.

The rest of the paper is organized as follows: Section II provides the background and motivation for our work. Section III presents our approach of dysplasia classification in IBD patients. We report the performance evaluation in Section IV. We provide a discussion in Section V and conclude in Section VI.

## II. BACKGROUND AND MOTIVATION

### A. DL for IBD

IBD clinically presents in two forms: ulcerative colitis and Crohn's disease. In recent years, DL has been explored for IBD diagnosis. Takenaka et al. [21] validated the effectiveness of a deep neural network for evaluation of endoscopic images of patients with ulcerative colitis. They achieved over 90% accuracy for identification of endoscopic remission and histologic remission. Stidham et al. [22] trained a CNN on

endoscopic images for endoscopic severity grading of ulcerative colitis. They showed that the CNN model could achieve similar performance to experienced human reviewers. Maeda et al. [23] developed a computed-aided diagnosis system to grade ulcerative colitis-related mucosal inflammation using endoscopy images. Their system used texture analysis of the images and machine learning to distinguish whether the histologic inflammatory status was healing or active. Kohli et al. [24] suggested that AI can be of great value for IBD due to subjectivity in the diagnosis using endoscopic evaluation.

Recently, Ho et al. [25] developed a DL approach for screening colorectal biospies for dysplasia, inflammation, and malignancy to diagnose colorectal cancer. They used WSIs and performed segmentation on the slides. Specifically, they used the Faster Region Based CNN architecture [26] for instance segmentation and ResNet [14] for feature extraction. The slide classification was performed using traditional machine learning to classify high-risk/low-risk slides.

Yamamoto et al. [27] developed a DL approach for classification of neoplasias in IBD patients using endoscopy images. They used EfficientNet [16] as the underlying CNN and showed that their model could achieve higher diagnostic accuracy than human experts. Similarly, Abdelrahim et al. [28] developed a DL model for neoplasia detection in IBD patients using endoscopic images and videos. They used the RetinaNet architecture [29] with ResNet for deep feature extraction.

Most of the prior efforts have employed DL on endoscopic images and videos of IBD patients.

### B. Bayesian DL

A Bayesian neural network is a neural network trained using Bayesian inference [19], [30]. Such a network enables the quantification of uncertainty in the predictions compared to traditional DL models that are typically considered black boxes and provide only point estimates. Furthermore, Bayesian DL is valuable in domains where limited data are available for training DL models [19]. A recent benchmarking study investigated the impact of different Bayesian DL techniques for diabetic retinopathy detection [31]. Significant improvement in classification accuracy and area under the curve (AUC) was achieved using different Bayesian DL techniques.

While there are different Bayesian DL techniques [19], [31], we are inspired by the work of Gal and Ghahramani [20], which introduced Monte Carlo dropout for Bayesian approximation. Their work showed that by adding dropout after every weight layer, approximately corresponds to variational inference (an approach for approximate inference) in a Bayesian neural network. Monte Carlo dropout is also easy to introduce in a deterministic DL model. During prediction/inference, dropout is enabled in a Bayesian DL model. As a result, the model is run several times (i.e., forward pass) to output a distribution of predictions on a test input. The mode (or mean) of the predictions can be used as the final prediction for classification (or regression). In contrast, a deterministic DL model produces only one prediction for the test input. Hence, a Bayesian DL model is typically slower for prediction.

## C. Motivation

Our work is motivated by two reasons: Firstly, most of the prior work focused on using endoscopic images/video for assessing the severity of IBD. Some of the approaches focused on neoplasia instead of dysplasia in IBD patients or dysplasia in colorectal cancer patients. Although, dysplasia is common to both IBD and non-IBD associated colorectal cancer, the dysplasia associated with IBD is more challenging to identify for a pathologist compared to non-IBD associated dysplasia. This is because the intensity of background inflammation in IBD associated dysplasia is higher and can be morphologically complex, thus posing significant challenge in identifying it. On the other hand, non-IBD associated dysplasia is easy to identify and characterize. Secondly, Bayesian DL can enable higher classification accuracy when data are limited. Recall that if dysplasia is missed or incorrectly graded, it can eventually lead to colorectal cancer in IBD patients. Thus, the benefit of Bayesian DL may outweigh the slower prediction time in a non-real time diagnostic application such as dysplasia classification of IBD patients.

## III. Our Approach

In this section, we introduce the different types of dysplasia of interest to a pathologist and then present our approach for dysplasia classification using Bayesian DL. (Our idea was first reported in a patent application [32].)

### A. Dysplasia Types for Classification

The three different types of dysplasia that we aim to accurately classify in WSIs of IBD patients are: (a) high-grade dysplasia (HGD), (b) low-grade dysplasia (LGD), and (c) no dysplasia (NEG). Figure 1 shows examples of HGD, LGD, and NEG patches extracted from WSIs.

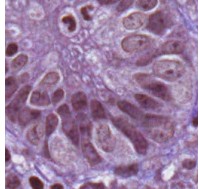 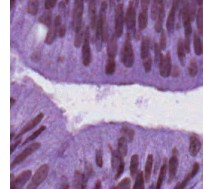 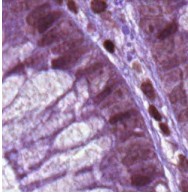

(a) An HGD patch    (b) An LGD patch    (c) A NEG patch

Fig. 1. Examples of different types of dysplasia

HGD involves severe dysplasia and may become cancer. The colonic crypts are proliferative, crowded, and hypercellular, exhibiting cribriforming, solid nests, and intraluminal necrosis. Histologically, HGD shows significant pleomorphism with cytologically malignant cells, rounded and heaped-up cells, and an increased nuclear-to-cytoplasmic ratio. Architectural complexity is evident with irregular, back-to-back tubules. Mitoses are increased and atypical.

On the other hand, LGD is characterized by mild to moderate epithelial dysplasia. The colonic crypts are proliferative, crowded, hypercellular, and arranged in parallel without architectural complexity. Histologically, LGD displays a

*picket fence* appearance with elongated, hyperchromatic, cigar-shaped, pseudostratified nuclei. There is varying maturation and mucin production with the presence of dystrophic goblet cells. The nuclei maintain a basal orientation, occupying the bottom half of the cell. Importantly, LGD does not exhibit atypical mitoses, loss of polarity, pleomorphism, or features such as back-to-back, cribriform, or budding tubules.

Finally, NEG shows a well-organized and orderly structure composed of a single layer of columnar epithelial cells with interspersed goblet cells. The crypts are lined with uniform, basally oriented nuclei, which occupy the basal third of the cell, and the cells exhibit a high degree of maturation as they migrate towards the luminal surface. The overall architecture lacks any evidence of dysplasia, pleomorphism, or abnormal mitotic activity.

In essence, classifying dysplasia is a challenging task and requires an expert pathologist with several years of training. Not all surgical pathologists have training in gastrointestinal pathology and therefore, may miss subtle dysplasia patterns. In this work, we investigate the potential of DL for classification of dysplasia in tissue sections of IBD patients. As a result, a pathologist can be assisted during diagnosis to avoid missing or misclassifying dysplasia.

### B. Dysplasia Classification Using Bayesian DL

Next, we present our approach for dysplasia classification in IBD patients. The overall steps in our approach are illustrated in Figure 2.

An expert pathologist identified IBD patients in the University of Missouri (MU) hospital (IRB No. 2070142) using NLP searches of pathology records for ulcerative colitis and dysplasia. The glass slides of these patients were first reviewed by the pathologist. A subset of these slides were selected and digitized using a WSI scanner. Next, the pathologist used Aperio ImageScope[4] to annotate rectangular regions in these slides for HGD, LGD, and NEG. The annotations were saved as XML files along with the original WSIs. The second expert pathologist reviewed only the annotations of the first expert pathologist to either confirm or reject the type of dysplasia that was identified. Only annotations that both pathologists agreed were retained for further processing.

As the annotations (i.e., rectangular regions) can be of different sizes and contain tens of thousands of pixels, they cannot be directly fed to a DL model that typically requires fixed-size images/image patches. Therefore, for each rectangular region, patches of a specific size (e.g., 256×256 pixels) were randomly extracted from inside of the region to create a dataset containing the three types of dysplasia. The OpenSlide[5] library was used to read the WSIs and extract the patches. The patches were then randomly split into training/validation set (80%) and testing set (20%).

Our next step is to introduce Bayesian DL into well-known DL models for image classification. The authors of Monte Carlo dropout [20] suggested adding dropout after every

---

[4] www.leicabiosystems.com/us/digital-pathology/manage/aperio-imagescope
[5] https://openslide.org

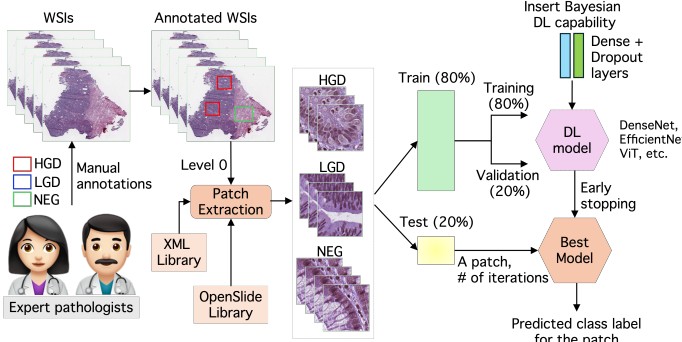

Fig. 2. Overall steps for dysplasia classification

weight layer in a neural network for Bayesian approximation. However, to avoid significantly increasing the number of parameters of a DL model that can have 100+ layers, we propose to insert one or more densely connected layers before the final densely connected layer of a model that performs classification. Each newly inserted densely connected layer is immediately followed by a dropout layer. These newly inserted dropout layers are enabled during training and inference. As a representative case, Figure 3 shows how we modified the DenseNet model to perform Bayesian DL using Keras. Specifically, the `add_layers()` function inserts the densely connected and dropout layers. Different dropout values can be used. Also, the number of neurons in the densely connected layers can be changed along with the activation function. (Similar modifications can be done with other popular frameworks such as PyTorch.) In our evaluation, we demonstrate that this modification to enable Bayesian DL is effective in improving the model performance for dysplasia classification, wherein very high accuracy is desired.

```
def DenseNet(....)
...
...
  x = layers.GlobalAveragePooling2D()(x)

  # Insert Bayesian deep learning capability
  b_dropout = ...
  b_neurons = ...
  b_levels = ...
  x = add_layers(layers, x, b_dropout, b_neurons, b_levels)

  x = layers.Dense(classes, activation=classifier_activation)(x)
...
...
def add_layers(layers, x, dropout, num_neurons, num_levels):
  for i in range(0, num_levels):
    x = layers.Dense(num_neurons, activation='relu')(x)
    x = layers.Dropout(dropout)(x, training=True)

  return x
```

Fig. 3. Modified DenseNet implementation in Keras for Bayesian DL

In Keras, when `training=False` in Dropout(), the

## TABLE I
### # OF PATCHES USED FOR TRAINING/VALIDATION AND TESTING

| # of patches for training/validation (80%) | | | | # of patches for testing (20%) | | | |
|---|---|---|---|---|---|---|---|
| Total | HGD | LGD | NEG | Total | HGD | LGD | NEG |
| 29,548 | 6,852 | 8,284 | 14,412 | 7,385 | 1,712 | 2,070 | 3,603 |

dropout layer is enabled only during training. This introduces regularization to prevent a model from overfitting [33] and perform better on the test set. However, when `training=True` as shown in Figure 3, the dropout layer is enabled during both training and testing. Hence, during inference, the modified model executes in forward pass for some number of iterations producing a distribution of predictions for a test input. The final predicted class label is the most frequent class label output by the model. The variance of the predictions can be used to understand the model's uncertainty. Thus, we aim to obtain more robust predictions for classifying dysplasia using the modified model rather than using its deterministic counterpart. While Bayesian DL increases the prediction time, the improvement in classification accuracy may outweigh the increased prediction cost due to the non-real time nature of the diagnosis.

## IV. PERFORMANCE EVALUATION

In this section, we report the evaluation of our approach for dysplasia classification using Bayesian DL on WSIs of histopathology tissue slides of IBD patients.

### A. Implementation and Setup

We implemented our software using Python (v3.8), Keras (v2.4.3), TensorFlow (GPU) (v2.4.1), and OpenSlide (v4.0.0). We ran all experiments on a server with Intel Xeon processor W-2245, 128 GB RAM, 1 TB solid state drive, and two Nvidia RTX A4000 (16 GB) GPUs.

We used three well-known DL models: DenseNet121, EfficientNetB0, and ViT. For DenseNet121 and EfficientNetB0, we used the default learning rate of 1e-3, the Adam optimizer, and the sparse categorical cross entropy loss function. For ViT, we used a learning rate of 1e-4, 64 transformer blocks, patch size of $32\times32$, and multihead attention with 4 attention heads. The AdamW optimizer was used with weight decay of 1e-4. As before, the sparse categorical cross entropy loss function was also used for ViT. Data augmentation was kept simple; random left/right flip was applied to improve the model performance. All models were trained using batch size of 32 for 1,000 epochs with early stopping. (The model weights were initialized with random values.) Note that we tried different hyperparameters; however, the above values gave the best results for the tested models.

For Bayesian DL, we used `b_neurons=1024`, `b_levels=1`, and `b_dropout=0.3` for all the models. Hereinafter, we will refer to the modified models as DenseNet121★, EfficientNetB0★, and ViT★. During training, the model that achieved the best validation accuracy was saved

| Model | Total # of parameters | Total # of correct predictions | Accuracy (%) | | | Sensitivity (%) | | | Specificity (%) | | |
|---|---|---|---|---|---|---|---|---|---|---|---|
| | | | HGD | LGD | NEG | HGD | LGD | NEG | HGD | LGD | NEG |
| DenseNet121 | 7,040,579 | 7,073 | 96.787 | 89.661 | 98.806 | 96.787 | 96.016 | 95.187 | 99.030 | 96.339 | 98.820 |
| DenseNet121★ | 8,090,179 | 7,103 | **97.371** | 91.884 | 98.084 | 97.828 | 94.862 | 96.137 | **99.208** | 96.858 | 98.139 |
| EfficientNetB0 | 4,053,414 | 7,124 | **97.371** | 91.207 | **99.056** | 97.257 | **96.721** | 95.966 | 99.206 | 96.917 | **99.072** |
| EfficientNetB0★ | 5,364,390 | **7,134** | **97.371** | **93.236** | 98.168 | **98.174** | 95.027 | **96.745** | **99.208** | **97.239** | 98.230 |
| ViT | 16,017,603 | 6,865 | 96.261 | 84.830 | 96.058 | 90.499 | 92.323 | 94.511 | 98.849 | 94.273 | 96.185 |
| ViT★ | 17,067,203 | 6,905 | 94.334 | 86.715 | 97.002 | 93.460 | 92.716 | 93.926 | 98.285 | 94.953 | 97.052 |

and used for evaluationg the test set. The modified/Bayesian DL models were run for 11 iterations during inference on the test set. The final prediction for a test input in a modified model was the class that was the most frequent among the 11 predictions. Note that the deterministic counterparts provided only one prediction per test input.

### B. Dataset

We used 61 de-identified WSIs of IBD patients (48 cases) for the evaluation. From each rectangular region/annotation, we extracted 10 random patches of size 256×256 pixels. Table I shows the number of patches in each class (and the total) for training/validation and testing.

### C. Classification Results

Next, we report the performance of the Bayesian DL models and their deterministic counterparts on the test set shown in Table I. Table II reports the classification accuracy, sensitivity, and specificity for the three classes. The total number of correct predictions for each model is also reported. Among the different models, EfficientNetB0★ performed the best followed by EfficientNetB0. Across all the models, we observed an improvement in the total number of correct predictions when Bayesian DL was used. This demonstrates the value of Bayesian DL for dysplasia classification in IBD patients, wherein high level of accuracy is desired.

### D. Training and Inference Time

Table II shows the number of parameters for each model. As observed, the modified models had an increase in the number of parameters (about 1M). As expected, the modified models were slightly slower to train compared to the original models. For example, DenseNet121★ was 3 minutes slower to train than DenseNet121. The inference time increased based on the number of iterations used for the modified models. For example, EfficientNetB0 required 1 min 13 sec to predict all the 7,385 patches in the test set. However, EfficientNetB0★ required 6 min 24 sec for the same test set using 11 iterations. We assert that the improvement in accuracy due to Bayesian DL outweighs the increased cost of prediction for diagnostic applications that are not real-time in nature.

### E. Impact of Image Patch Size

We tested the models on a dataset created with a smaller patch size of 128×128 pixels. However, this deteriorated the LGD classification accuracy significantly and was below 80% for all the models. After consultation with an expert pathologist, it became evident that LGD detection pattern is complex and requires the observation of colonic crypts rather than a few cells for accurate detection. Hence, a larger patch size of to 256×256 pixels was more effective for dysplasia classification on the tested WSIs.

## V. DISCUSSION

IBD-associated dysplasia is harder to identify and characterize compared to other types of dysplasia. If incorrectly classified, it could lead to colorectal cancer in IBD patients. To the best of our knowledge, our work is the first to show how Bayesian deep learning can be applied for accurate dysplasia classification in IBD patients using whole slide images. Hence, we cannot directly compare with prior work on dysplasia classification. However, we use well-known deep learning models to test the generality of our solution.

We introduced Monte Carlo dropout [20] so that it is active during training and inference to enable Bayesian DL. Rather than changing the dropout layers of existing models, we inserted densely connected layers with dropout to eventually feed into the final fully connected layer. This way we did not change the core architecture of well-known models but introduced the required randomness via dropout during training and inference. Our approach is simple yet effective in achieving good performance. Although our approach does not always improve the classification accuracy of individual classes, it does improve the overall classification accuracy of all the three well-known models (see Table II). For example, while EfficientNetB0 correctly classified 7,124 patches, EfficientNetB0★ correctly classified 7,134 patches.

We tested the original models (e.g., EfficientNetB0) and the modified ones (i.e., EfficientNetB0★). This can be considered as an ablation study wherein we tested the impact of adding the densely connected and dropout layers to the original models (i.e., removing them).

## VI. Conclusion

We proposed a Bayesian DL approach for accurate classification of dysplasia in IBD patients using WSIs as IBD-associated dysplasia is challenging to detect compared to other types of dysplasia. We modified well-known DL models for image classification by inserting one or more densely connected layers before the final densely connected layer of a model that performs classification. Each newly inserted layer is followed by a dropout layer. These inserted dropout layers are enabled during training and inference. As a result, the core architecture of a chosen model is not changed. During inference, a distribution of predictions is obtained, and the most probable class label is computed for a test input. Using 60+ WSIs, we achieved the best patch-wise classification accuracy using EfficientNetB0⋆, which achieved an overall accuracy of 96.6% (considering all classes). In general, LGD was more challenging to classify for all the models due to its complex nature. We also observed that Bayesian DL models achieved better performance than their deterministic counterparts albeit increase in prediction time. In the future, we plan to obtain independent annotations from multiple pathologists to evaluate our model to assess the interobserver agreement. We also plan to leverage parallelism to reduce the inference time using Bayesian DL.

## Acknowledgments

This work was funded by a grant awarded to Deepthi Rao and Praveen Rao by the MU Coulter Biomedical Accelerator Program. We thank the anonymous reviewers for their insightful comments and suggestions.

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
