# OpenReview forum: "Accurate Classification of Dysplasia in Inflammatory Bowel Disease Patients Using Deep Learning"
_IEEE.org/EMBS/BHI/2024/Conference — IEEE BHI'24_

### Official Review · Reviewer_8nHy · 2024-08-05
**Accurate Classification of Dysplasia in Inflammatory Bowel Disease Patients Using Deep Learning**

**Overall Rating:** 6
**Confidence:** 4

**Other Quality Metrics:**

(a) Clarity of writing: fair
(b) Clinical Significance: fair
(c) Methodological Novelty: poor
(d) Experiments and Results: good

**Questions For The Authors:**

(1) Could authors explain why they need to insert one or more densely connected layers?
(2) Could authors explain the novelty of the model?
(3) Could authors do the ablation test?

**Strengths:**

(1) Compared with the SOTA model and improved the classification accuracy.

**Summary Of The Paper:**

The authors develop an approach for accurate classification of dysplasia in IBD patients using Bayesian deep learning.  The existing deep learning models are modified to perform Bayesian approximation for achieving higher classification accuracy than a deterministic deep learning model.

**Weaknesses:**

(1) The novelty of the model.
(2) Smaller dataset.

---

### Official Review · Reviewer_ixbD · 2024-08-12
**Classification of Dysplasia in Inflammatory Bowel Disease**

**Overall Rating:** 7
**Confidence:** 4

**Other Quality Metrics:**

(a) Clarity of writing;great
(b) Clinical Significance; excellent
(c) Methodological Novelty; excellent
(d) Experiments and Results; great

**Questions For The Authors:**

I do not have any questions

**Strengths:**

The authors make a strong argument for the importance of their work. Automated approaches would be very beneficial for the early detection of pre-cancerous histopathological changes and therefore for the prevention of cancer in IBD patients.

**Summary Of The Paper:**

The aim of the paper is to improve ML-based detection of specific histopathological changes in cells of patients with inflammatory bowel disease. Specifically, dysplasia is to be detected in microscopic slides using deep learning. The authors describe a modification of an established deep-learning approach.

**Weaknesses:**

none noted

---

### Official Review · Reviewer_ddUT · 2024-08-15
**Improving dysplasia detection in IBD patients through advanced deep learning techniques**

**Overall Rating:** 7
**Confidence:** 5

**Other Quality Metrics:**

a) clarity of writing: great
b) clinical significance: great
c) methodological novelty: good
d) experiment and results: good

**Questions For The Authors:**

-The paper can benefit from discussion as to why the three different models with their Bayesian approximation result is different result and in some the performance is even worst than the model without Bayesian approximation.
-The paper would benefit from comparison to other method in the filed and how their approach in helping increase the accuracy and to what percentage and what are the implication of it.
-It would be good to discuss in this kind of decisions like what is the optimization for, what is the threshold for missing a case or giving a false alert to someone, sensitivity vs specificity in disease detection.
-In annotation from the pathologist, would be good to clarify whether both were blind to each others diagnosis or the next pathologist only could confirm or reject the first one's diagnosis.

**Strengths:**

The paper is clear and well-organized, making it easy to understand the authors' ideas and findings.
The problem they are proposing is an import one in healthcare. Detecting dysplasia sooner and getting treatment can reduce the mortality rate in IBD patients.
The author's approach in choosing successful deep learning model and their bayesian approximation to it is novel and interesting approach.

**Summary Of The Paper:**

The inherent difficulty of dysplasia classification is amplified in IBD, underscoring the need for advanced diagnostic approaches that can overcome the challenges posed by inflammation in these patients. In this paper, the authors proposed a new method using Bayesian approximation in three different deep learning models to increase the accuracy of detections. Their proposed models achieved accuracy of 97.23%, 93.23% and 98.16% respectively.

**Weaknesses:**

The paper is lacking a comprehensive discussion and conclusion section. Topics like 1) how the results of their new approach is better than other published ones, to what percentage, what is the reason behind choosing "accurate classification .." as title. 2) why adding Bayesian approximation is not enhancing all three of their suggested models.

---

### Decision · Program_Chairs · 2024-09-23

Accept